# An Application of Partial Least Squares Structural Equation Modeling (PLS-SEM) to Examining Farmers' Behavioral Attitude and Intention towards Conservation Agriculture in Bangladesh

**Riffat Ara Zannat Tama** [1,2], **Md Mahmudul Hoque** [3], **Ying Liu** [4,*], **Mohammad Jahangir Alam** [5]
**and Mark Yu** [6,*]

1    Department of Agricultural Economics, Bangladesh Agricultural University, Mymensing 2202, Bangladesh
2    College of Economics and Management, Huazhong Agricultural University, Wuhan 430070, China
3    Ministry of Public Administration, Dhaka 1000, Bangladesh
4    College of Economics and Management, North China University of Technology, Beijing 100144, China
5    Department of Agribusiness and Marketing, Bangladesh Agricultural University,
     Mymensing 2202, Bangladesh
6    Division of Agribusiness and Agricultural Economics, Department of Agricultural and Consumer Science,
     Tarleton State University, P.O. Box T-0040, Stephenville, TX 76402, USA
*    Correspondence: liuying7223@163.com (Y.L.); yu@tarleton.edu (M.Y.)

**Abstract:** Despite being widely advocated as a climate-smart farming system, the adoption of conservation agriculture (CA) among Bangladeshi farmers has remained surprisingly low. Evidence indicates that farmers' behavior regarding the adoption and continuation of CA is affected by their socioeconomic and psychological factors. This study combined the Technology Acceptance Model (TAM) and Diffusion of Innovation (DOI) theories to examine the socio-psychological determinants of Bangladeshi farmers' behavior regarding the adoption of CA. The proposed model included both reflective and formative measurements. Based on data collected from 201 CA farmers, this research used a variance-based structural equation modeling (PLS-SEM) approach to test the model. The analysis showed that the components of this integrated model explained more variance (Intention: 48.9%; Attitude: 59.2%) than the original TAM framework (Intention: 45.8%; Attitude: 54.5%). Farmers' attitudes toward the continuation of CA were most influenced by the Relative Advantage (RA) of CA ($\beta = 0.337$). The low level of Complexity ($\beta = 0.225$) and Compatibility ($\beta = 0.273$) of CA had a significant positive effect on attitude. In a campaign to encourage farmers to act more sustainably, interventions should emphasize CA's long-term benefits, such as its effects on soil, yield, and the environment.

**Keywords:** conservation agriculture; attitude; intention; behavioral sustainability; Bangladesh





## 1. Introduction

The world population will reach 9 billion by 2050, a 15% increase from 7.8 billion by 2020, which challenges both availability and access to food [1]. Rapid population growth and industrialization cause environmental degradation and land fragmentation, which in turn exert pressure on the climate with growing greenhouse gas emissions. Consequently, a significant portion of the global population is on the verge of food insecurity amid the threat posed by climate change [2,3]. The situation is exacerbated by scant climate change adaptation strategies, which result in a lower food supply [4]. This necessitates changes in the agricultural food production system toward one that is more sustainable, efficient, and reliable to ensure global food security under the pressure of a growing population and climate change [5–7]. The FAO [8] argues that agricultural food production can be

increased by 70% by applying scientifically sound, socially acceptable, and environmentally friendly farming practices.

In the epoch of modernization and technological advancement, when climate change has become a major global development concern, conservation agriculture (CA) can play a pivotal role in ensuring food supply while preserving the Earth. CA is a set of principles that guide the adoption of reliable, sustainable, and climate-resilient farming practices [9]. Three principles of CA are: (i) minimum or no tillage and minimum soil disturbance; (ii) permanent soil cover, with crop residue or a live cover crop; and (iii) crop diversification, rotation, and intercropping (different crops alternated in the same field and preferably cereals followed by legumes) [10]. Dumansky et al. [11] found that existing deep tillage-based conventional farming (CF) posed a great threat to the quality depletion of natural resources, including soil, topography, water, and biodiversity. In contrast, minimum soil cultivation in CA decreases the necessity for agricultural machinery input, which in turn reduces overall cultivation costs [12]. CA is a proven resource-effective farming system in which crops are cultivated with minimum soil disturbance and no burning of residue, and can preserve natural biodiversity through minimum tillage [8]. Tillage of CA and application of optimum irrigation water increase soil nutrient cycling and organic matter, which paves the way for the efficient use of irrigation water [13,14]. Alongside, CA ensures environmental and economic sustainability by reducing the demand for inputs such as seed, fuel, water, and fertilizer [15,16]. According to Jat et al. [17], CA is adaptive to extreme climatic conditions of farming, which can increase crop yields and mitigate negative environmental effects, such as water stress and terminal heat. Evidence shows that CA is remarkably beneficial for agricultural land in Bangladesh [14,18–21].

However, evidence indicates that in the early years, the adoption of CA can reduce yields [22]. Around the globe, not many (8–15 percent) farmers have adopted the CA practice [5,23–25]. In a country such as Bangladesh, which is among the worst victims of ongoing climate change, CA practices can help mitigate environmental degradation and safeguard its ecological balance. CA was introduced in Bangladesh in the 1990s. Despite its immense potential, the adoption of CA in Bangladesh remains low (for instance, 6.6% among maize farmers) and unsustainable [26]. In 2015/16, the land coverage of CA in Bangladesh was 1500 hectares [27]. However, there is a lack of survey data regarding the current trend of CA adoption in Bangladesh. In the race to achieve the SDGs by 2030, CA practice can be a prudent approach to achieving SDG 2 (zero hunger and sustainable agriculture) and SDG 13 (combating climate change). Previous studies have mostly focused on socioeconomic, technological, and environmental factors and issues regarding CA adoption [21,28–30]. Hence, it is pertinent to explore the issues related to CA practice and unpack what prevents the adoption and continuation of CA practice. This study set out with the following questions: (i) Why are CA farmers not keen to continue CA farming practices? What are the factors that influence farmers' attitudes toward the continuance intention of CA applications in Bangladesh? To answer these questions, this study combines the variables hypothesized by the Diffusion of Innovation (DOI) and Technology Acceptance Model (TAM) theories into a single model.

Fishbein and Ajzen [31] theorized that the behavior of a human individual is directly related to his or her behavioral intention. According to the TAM, behavioral intention is influenced by attitude toward the technology or innovation [32]. Attitude is the main driver of behavioral intention [33–35]. According to DOI theory, the adoption of technology or innovation is not a straightforward process [36]. The adoption rate is higher if an individual identifies a technology as having more relative advantage, compatibility, observability, trialability, and less complexity. In support of this theory, a plethora of studies have claimed that Relative Advantage (RA), Compatibility, and Low Complexity (LC) levels increase the likelihood of adoption [37–39]. Therefore, this study assumed RA, LC level, compatibility from DOI, and attitude from TAM to explain farmers' behavioral intention toward continuing CA practice in Bangladesh.

Recent studies have applied TAM to examine behavior and behavioral intention in different contexts [40–42], and several pieces of literature [43,44] have utilized this theory to assess attitude, intention, and behavior in relation to CA. Many researchers have used DOI theory in context-specific studies, some of which have a particular focus on the area of conservation [14,45–48].

Similar to this study, several studies have previously integrated these two theories: TAM and DOI [49–51]. Such integration has been used successfully to explain behavior's adoption of new technologies [52]. Gefen and Straub [53] and Koufaris [54] applied these theories to study e-commerce, while Aldás-Manzano et al. [55], Lu et al. [56], and Luarn and Lin [57] applied them to banking research on cell phone technologies. Koenig-Lewis et al. [38] notably combined TAM and DOI theories to predict younger consumers' behavior toward mobile banking. Being stimulated by this integration, this study contributes to the body of literature by assessing the underlying socio-psychological factors that construct an individual's attitude toward the behavioral intention of continuing CA practice.

The overall objective of this study was to understand the behavioral intentions of CA farmers by dissecting their attitudes toward it. More specifically, this research measures the impact of the components of attitudes on farmers' intentions.

The discussion above highlights two crucial aspects of the current scholarship on CA. First, the issue of the sustainability of farmers' behavior in developing nations was identified through adoption-related research. While some smallholder farmers tend to partially adopt CA, others may not continue with the technique [58,59]. However, few studies have investigated the underlying psychological factors behind farmers' mysterious and unsustainable behaviors. Second, the theoretical literature indicates that attitude remains a major factor in farmers' intentions to continue novel technological adoption. However, the existing empirical literature lacks evidence regarding farmers' attitudes toward continuing CA. This research contributes to this knowledge gap by focusing on farmers' attitudes in Bangladesh, where previous research has identified farmers' unsustainability of adoption-related behavior [26,58].

To the authors' knowledge, this is the very first empirical study concerning CA practice in Bangladesh that uncovers farmers' behavioral intentions toward CA practice and, at the same time, uniquely integrates the DOI and TAM theories to scrutinize the psychological driving factors influencing farmers' intentions to continue this practice. Moreover, our study contributes to the existing literature in several ways. First, it has opened a new avenue for research on CA practice in Bangladesh. Second, the study has shown whether, in a developing country such as Bangladesh, the TAM theory and the DOI theory for any new technological adoption hold true. In Bangladesh, most farmers and smallholders (in terms of land size and capital) have unique and extra risks due to their vulnerable socio-economic status. Third, it offers guidance for exploring the potential of CA practice in Bangladesh. This article is organized as follows: Section 1 is followed by the conceptual framework and methods sections. The subsequent sections outline the results, discussion, and conclusions.

## 2. Conceptual Framework

DOI theory and TAM theory are two widely used theories in the study of innovation adaptation [60]. The DOI theory, which was proposed by Rogers [61], can be used to explain the factors influencing the decision to adopt an innovation. Multi-disciplinary studies [62–65] used DOI from various perspectives, whereas the TAM emphasizes the intention of an individual regarding a technology adoption. These studies indicate that DOI and TAM are considered ideal for complementary uses because of the common conceptual premises they share. However, some studies considered TAM to be a part of DOI and showed that the predictability of TAM can be enhanced if an integrated model is used, combining TAM with DOI [66]. Adding other innovation characteristics [37,67] can also help TAM make better predictions [32,67].

*2.1. The Theory of the Technology Acceptance Model*

The Theory of Technology Acceptance Model (TAM) was introduced and developed by Davis in 1989 as a modification of the Theory of Reasoned Action by Fishbein and Ajzen [31]. This theory consists of two fundamental components, namely Perceived Usefulness (PU) and Perceived Ease of Use (PEOU). Several cross-disciplinary studies have effectively used TAM to predict user acceptance [56,68–71]. It has also been extensively applied in the area of agricultural technology adoption [72–75]. TAM has five constructs, namely Perceived Usefulness (PU), Perceived Ease of Use (PEOU), Behavioral Intention of Use (BIU), Attitude toward Using (ATU), and Actual System Use (AU). PU and PEOU can be used to assess an individual's attitude toward a particular technology. PU is the degree to which an individual believes that using a specific technology improves practice performance over existing technology [76]. On the one hand, PU has a positive influence on a person's intention and attitude [77–80]. On the other hand, PEOU denotes the technology that an individual finds easy to use and requires little physical and mental effort. Venkatesh and Davis [81] showed that technology and innovation are more useful when they are easier to use. Schuitema et al. [82] revealed that the measurement of actual adoption is hard to calculate. The construct of RA in DOI is closely related to PU in TAM, whereas PEOU in TAM has similarities with the complexity in DOI [38]. To improve TAM predictability, additional variables should be added to existing TAM constructs [83]. Table 1 shows the measurement scales for the constructs and previous studies associated with these constructs.

**Table 1.** Measurement scale of the construct.

| Construct | Scale | Number of Items | Sources |
|---|---|---|---|
| Intention | Reflective | 5 | [33,84–86] |
| Attitude | Reflective | 3 | [33,68,85,86] |
| Relative Advantage (RA) | Reflective | 4 | [68,85,87] |
| Low Complexity (LC) level | Formative | 3 | [33,68,85,87] |
| Compatibility (COMP) | Formative | 3 | [63–84] |

Attitude is associated with positive or negative feelings when executing any behavior. Therefore, an individual's attitude is their psychological inclination to form a favorable or negative judgment of their conduct [58,88–90]. According to TAM, attitude is influenced by both PU (which is described as RA in the study) and PEOU (which is labeled as the Low Complexity level in the study). Davis [76] argued that behavioral intention can be more vividly forecasted than actual behavior. Ajzen [91] also indicated that the actual behavior could be accurately predicted when the intention of that behavior was evaluated. Therefore, analyzing intention is crucial for predicting actual behavior. Following this line of thought, this study focused on comprehending behavioral intention.

*2.2. The Theory of Diffusion of Innovation*

The rate or level of innovation adoption is based on an individual's perception of innovation attributes. As per the Theory of Diffusion of Innovation (DOI), these innovation attributes comprise five characteristics: Relative Advantage, Complexity, Compatibility, Observability, and Trialability, by which 49–87% of the variation in the adoption rate can be explained [61].

Relative Advantage refers to the extent to which a novel technology is deemed more advantageous than the existing ones, which can be evaluated by economic benefits, satisfaction, suitability, social aspects, and prestige [92,93]. Adoption rates are expected to rise if early adopters benefit significantly more than the traditional practice, and the adopter believes that the new technology offers a greater relative advantage. The term "complexity" refers to the context in which an invention or technology is difficult to comprehend and utilize, resulting in a lower adoption rate. Adopting an innovation requires new skills and a broader understanding of the technology. Beyene and Kassie [94] revealed that the adoption rate is influenced by the skills acquired by younger and beginning farmers when

implementing a new practice. The relationship between complexity and adoption rate is negatively correlated, and complexity contradicts other attributes. However, in this study, complexity is defined as the Low Complexity (LC) level, which is positively linked to attitude. Compatibility is the consistency of an innovation with an individual's existing values, practices, beliefs, and experiences. Reimer et al. [95] noted that farmers would reject a new technology if it was found to be incompatible with their current practices. Observability refers to the degree to which the innovation result is visible to others, while trialability is the degree to which a new technology may be used or experimented with by an individual on a partial or limited basis [61]. Since the farmers who took part in this study had prior experience with CA farming, observability and trialability were not taken into account.

Therefore, this study integrates the two models (DOI and TAM) into a framework (Figure 1), which not only provides greater predictability than each individual model but also helps check out the cross-correlation of the predictive constructs. The study sets out the hypotheses, which are as follows:

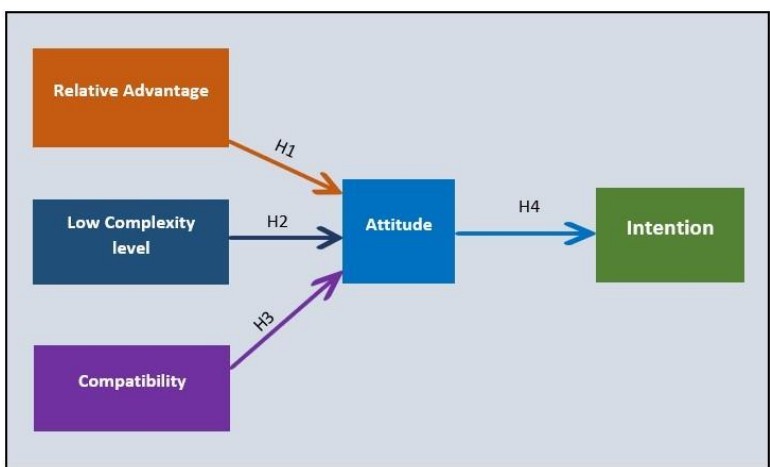

**Figure 1.** Conceptual framework.

## 3. Material and Methods

### 3.1. Survey Procedure

A sequential, multi-stage sampling approach was implemented to determine the targeted region and homes for this study. Arriagada et al. [96] noted that mixed methods (sequential) are widely used in studies related to agriculture, addressing a complex phenomenon, such as farmers' behavior. Data collection can be divided into several stages. In the initial stage, three northern districts were selected (Rajshahi, Rangpur, and Dinajpur) where CA-based tillage technology of crop production has been dominant [97]. Several national and international projects have been operating for years in this region to familiarize CA with the farmers [98]. Farmers in this region are well aware of the practice. These districts lie in a drought-prone zone, which makes irrigation either expensive or not readily available [99]. As a result, farmers can adopt CA, which will improve the efficiency of the use of limited water resources.

In the second stage, nine villages from five upazilas (Charghat, Godagari, Durgapur, Birganj, Pirganj, and Gangachara) of these three districts were selected. In the third stage, an accumulated list of farmers was obtained from the Regional Agricultural Research Station (RARS) and the Regional Wheat Research Center (RWRC). The research method's adequacy and reliability were tested in a pilot study with thirty (30) farmers. Between July and September 2019, a total of 220 farmers were recruited following Krejcie and Morgan's [100] approach of selecting the smallest sample size. During direct interviews, we obtained CA-related information from several families. Due to non-response from participants and some incomplete information, the effective sample size was reduced to 201. According

to Marcoulides and Saunders [101], the predicted adequate sample size can be calculated based on the maximum number of arrows pointing at a latent variable; the minimum sample size for our model is 70, as the model contains 5 arrows, which is the maximum number of arrows pointing at the latent variable (Intention) in the model. When testing the proposed model, a sample of 201 respondents was sufficient. Farmers were considered samples if they had prior experience with CA farming for at least one year on their land. All respondents were the heads of their homes and were in charge of more than 70% of their households' agricultural activities.

Finally, with the support of the respondents, this study conducted nine focus group discussions to extract complementary information. Following this, the data were examined and prepared for further investigation.

To collect information from the selected respondents, a survey instrument (structured questionnaire) was formulated. The questionnaire had two parts: the first part gathered information about their socioeconomic factors, such as age, education level, farm size, household size, and experience with CA; cropping pattern; annual income; and so forth, while the second part encompassed factors related to farmers' attitudes and their behavioral intentions toward CA (i.e., Intention, Attitude, Relative Advantage (RA), Complexity, and Compatibility). A number of activities, including a literature review, group discussions with agricultural experts and researchers from reputable institutions, and a pilot study, were used to complete the survey questionnaire. Following Rezaei et al. [44], this study noted respondents' responses on a Likert scale ranging from 1 (strongly disagree) to 5 (strongly agree). The optimal number of items or indicators per construct was three to five, as recommended by Bagozzi and Heatherton [102] (Table 1). Further, with the consent of the respondents, enumerators conducted face-to-face, questionnaire-guided interviews. Before being reviewed further, the majority of the quantitative data were first gathered in the local unit and then transformed into the standard unit.

### 3.2. Analytical Methods

Structural Equation Modeling (SEM) was used to test the hypotheses (Table 2). SEM approaches have two applications: variance-based techniques and co-variance-based techniques. Covariance-based SEM (CB-SEM) mainly deals with dimensions that are reflective in nature and does not deal with formative measurement models [81]. Jarvis et al. [103] conducted a systematic literature review to highlight that indicators that were measured reflectively in selected studies should have been measured formatively. They added that it raised concerns about the results and their practical implications. To overcome this limitation, the PLS component-based algorithm is a widely used software application utilized in several multi-disciplinary studies [104–107].

**Table 2.** Research hypotheses.

| Code | Hypotheses |
|---|---|
| Hypothesis 1 (H$_1$) | Relative Advantage (RA) has a positive influence on attitudes towards CA farming. |
| Hypothesis 2 (H$_2$) | There is a positive relationship between the Less Complexity (LC) level and attitude. |
| Hypothesis 3 (H$_3$) | Compatibility (COMP) has a positive influence on attitude. |
| Hypothesis 3 (H$_4$) | Farmers' attitudes have a positive influence on their intentions toward CA farming. |

Because this study involves both formative and reflective indicators and measurement, the suggested model was validated using a variance-based SEM (PLS-SEM) technique. CB-SEM is more confirmatory in nature, whereas variance-based SEM (PLS-SEM) appeases sample size, data normality, and indicator number assumptions. On the other hand, the latter helps to form theory rather than test it [108,109], which is consistent with the objectives of this study.

As this study combines two theories and creates an integrated novel model for explaining behavioral continuation intention toward CA, it is also important to evaluate the model's predictive capacity instead of theory conformity. This study adopted the two-step approach proposed by Anderson and Gerbing [110]. First, the measurement model was evaluated by assessing its reliability and validity. Second, a structural model was analyzed by estimating the path coefficients and assessing the significance of the path relationships.

SmartPLS version 3.2.4 [111] was used to assess the significance of the factor loadings and path coefficients.

Robustness Checks in PLS-SEM

The dataset of this study was initially checked using the observational method (mean $\pm$ 3 standard deviations) for any extreme observations or outliers. No such outliers were reported. The robustness check in PLS-SEM is outlined below.

(A)   Non-linear effects

This study used a two-stage method, which was first suggested by Chin et al. [112], to measure quadratic effects. This was done to see if there was a significant non-linear relationship between the constructs.

In the first stage, the main effect PLS path model was run in order to obtain estimates for the latent variable scores. The latent variable scores were calculated and saved for further analysis.

In the second stage, a quadratic term was construed as the element-wise product of the latent variable scores of the exogenous variable. In a multiple linear regression, the scores of the latent variables and the quadratic terms were used as independent variables to explain the scores of the latent variables of the endogenous variables.

As Table 3 shows, all the quadratic effects were found to be insignificant, indicating that there was no serious non-linear relationship between the constructs. The size of the quadratic effect ($f^2$) also reflected similar results.

**Table 3.** Assessment of non-linear effects.

| Path | Coefficient | PCI | *p*-Value | $f^2$ |
|---|---|---|---|---|
| QE_ATTonINT | −0.065 | [−0.185; 0.051] | 0.281 | 0.004 |
| QE_LConATT | −0.023 | [−0.198; 0.019] | 0.090 | 0.005 |
| QE_RAonATT | 0.068 | [−0.059; 0.187] | 0.272 | 0.002 |
| QE_COMPonATT | 0.050 | [−0.029; 0.130] | 0.212 | 0.000 |

Note: The quadratic effects assessed are based on a two-tailed percentile bootstrapping test at a 5% confidence level [2.5%; 97.5%]. PCI denotes percentile confidence interval; QE denotes quadratic effect.

(B) Unobserved heterogeneity

To check for unobserved heterogeneity, the FIMIX-PLS procedure was run on the data. To determine the maximum number of segments to extract, the minimum sample size required to estimate each segment was first computed. Here, the minimum sample size for PLS-Sem in this study was reported at 70.

Furthermore, the data were divided into four segments. AIC3 and CAIC indicate the same number of segments. Sarstedt et al. [113] revealed that when AIC3 and CAIC indicate the same number of segments, the results likely point to the appropriate number of segments. The results of this study showed that both AIC3 and CAIC point to a two-segment solution (Table 4). AIC4 and Bayesian information criteria (BIC) generally performed well when used to determine the number of segments in FIMIX-PLS. Both criteria pointed to a two-segment solution, which appeared to be densely clustered according to the EN criterion. A two-segment solution also met the minimum sample size requirements for each segment. However, the minimum description length with factor 5 (MDL5) also pointed to a two-segment solution (Table 5). Therefore, it was assumed that unobserved heterogeneity was not at a critical level.

**Table 4.** Fit indices to assess heterogeneity.

| Criteria | Segment1 | Segment2 | Segment3 | Segment4 |
|---|---|---|---|---|
| AIC (Akaike's Information Criterion) | 953.7 | 807.551 | 842.638 | 800.921 |
| AIC3 (Modified AIC with Factor 3) | 959.7 | 827.551 | 855.638 | 827.921 |
| AIC4 (Modified AIC with Factor 4) | 965.7 | 847.551 | 868.638 | 854.921 |
| BIC (Bayesian Information Criteria) | 973.52 | 873.617 | 885.581 | 890.110 |
| CAIC (Consistent AIC) | 979.52 | 893.617 | 898.581 | 917.110 |
| HQ (Hannan Quinn Criterion) | 961.72 | 834.284 | 860.014 | 837.011 |
| MDL5 (Min Description Length with Factor 5) | 1280.799 | 1107.882 | 1161.353 | 1462.867 |
| LnL (LogLikelihood) | −470.85 | −383.776 | −408.319 | −373.46 |
| EN (Entropy Statistic (Normed)) | N/A | 0.641 | 0.545 | 0.567 |
| NFI (Non-Fuzzy Index) | N/A | 0.611 | 0.555 | 0.586 |
| NEC (Normalized Entropy Criterion) | N/A | 87.067 | 91.482 | 72.114 |

Note: N/A denotes not available. Numbers in bold indicate the best outcome per segment retention criterion.

**Table 5.** Relative segment sizes.

| No. of Segments | 1 | 2 | 3 | 4 |
|---|---|---|---|---|
| 1 | 1.000 | | | |
| 2 | 0.707 | 0.293 | | |
| 3 | 0.552 | 0.311 | 0.137 | |
| 4 | 0.540 | 0.301 | 0.102 | 0.057 |

(C) Endogeneity

To detect endogeneity concerns, this study used the Gaussian copula approach proposed by Park and Gupta [114], which controls for endogeneity by directly modeling the correlation between the endogenous variable and the error term by means of a copula. This study first verified whether the variables were normally distributed or not. This was done by running the Kolmogorov–Smirnov test with Lilliefors' correction [115]. By checking all Gaussian copulas included in the model, it was found that none were significant (Table A1). There was no evidence of endogeneity.

## 4. Results

### 4.1. Measurement Model Assessment

The conceptual model includes both formative and reflective types of measurement scales. Among the five constructs, two (Low Complexity Level and Compatibility) have formative measurement scales, while the other three (Intention, Attitude, and Relative Advantage) have reflective measurement scales. The statistical evaluation criteria for reflective and formative models differ [116]. Internal consistency was not considered for the formative measurement model, as the items of the model were independent in nature and not highly correlated with each other [116,117].

The assessment guidelines are summed up in Table 6, which is based on several relevant studies [86,118,119]. The outer loading of the reflective measurement model must be higher than 0.6, and the two should be correlated with each other (Table 6). The constructs of the reflective measurement model were tested for reliability and validity in this study. For formative measurement models, convergent validity, collinearity, and outer weights were assessed. Table 6 depicts the important criteria for reflective and formative measurement approaches to be acceptable.

**Table 6.** Assessment of measurement models.

| Criterion | Guideline |
|---|---|
| **Assessment of reflective measurement model** | |
| Composite Reliability (CR) | CR > 0.70 |
| Indicator Loadings | Outer loadings >0.60 |
| Average Variance Extracted (AVE) | AVE $\geq$ 0.50 |
| Fornell–Larcker Discriminant Validity | AVE should be higher than the highest squared correlation with any other construct |
| Heterotrait–Monotrait Ratio (HTMT) | Value should be smaller than 1 |
| Cross Loadings | The loadings of each indicator on its construct are higher than cross-loadings on other constructs |
| **Assessment of formative measurement model** | |
| Convergent Validity (Redundancy analysis) | $\geq$0.70 Correlation value |
| Collinearity assessment (VIF) | Ideal VIF value <3.3 |
| Outer weights | Should be statistically Significant |

### 4.2. Assessment of Reflective Measurement Models

Following the approaches used by Hair et al. [116] and Henseler et al. [119], all reflective constructs (Intention, Attitude, and Relative Advantage) were analyzed to evaluate their reliability and validity. To assess internal consistency, composite reliability and Cronbach's alpha from a measurement model were used. The average variance extracted (AVE) and outer factor loadings of the indicators were assessed to measure convergent validity. Table 7 shows the constructs with acceptable values for factor loadings: composite reliability (CR), Cronbach's alpha value, and AVE. Table 7 shows that all constructs' composite reliability and Cronbach's alpha are above the minimum acceptance level of 0.70, which indicates that the constructs are reliable.

**Table 7.** Construct internal consistency, reliability, and validity.

| Factor | Notation | Items | Factor Loading | Composite Reliability (CR) | Cronbach's Alpha | AVE |
|---|---|---|---|---|---|---|
| Continuance Intention | INT 1 | Continue the practice next year | 0.716 | 0.872 | 0.872 | 0.577 |
| | INT 2 | Adopt in the near future | 0.755 | | | |
| | INT 3 | Continue the practice by himself or herself | 0.755 | | | |
| | INT 4 | Interested in the practice for the betterment of future generations | 0.766 | | | |
| | INT 5 | Inspire friends, relatives, and neighbors to adopt the practice | 0.804 | | | |
| Attitude | Att 1 | Good for soil health | 0.690 | 0.783 | 0.782 | 0.547 |
| | Att 2 | Requires less input cost | 0.779 | | | |
| | Att 3 | Decreases pest infestation | 0.748 | | | |
| Relative Advantage | RA 1 | Higher return on investment | 0.806 | 0.812 | 0.814 | 0.523 |
| | RA 2 | Higher yield after certain period | 0.656 | | | |
| | RA 3 | Environmentally friendly approach | 0.615 | | | |
| | RA 4 | Less labor required | 0.796 | | | |
| Less Complexity | LC 1 | Less complex procedures | 0.827 | | | |
| | LC 2 | Machineries available on time | 0.691 | | | |
| | LC 3 | Availability of labor on time | 0.847 | | | |
| Compatibility | COMP 1 | Fits with social norm | 0.811 | | | |
| | COMP 2 | Takes less efforts | 0.822 | | | |
| | COMP 3 | Compatible with current practices | 0.862 | | | |

The Fornell–Larcker criterion, Henseler's heterotrait–monotrait (HTMT) criterion, and cross loadings were applied to evaluate discriminant validity. The results are presented in Tables 8–10. In the measurement model, all constructs were distinctively different because the Fornell–Lacker criterion values (Table 8 were the square root of the AVE, which were higher than the correlation values. Henseler's HTMT criterion stated that all constructs (Table 9) were different from each other, with a threshold value of 0.90 [119]. Therefore, there was no discriminant validity issue in the measurement model. Another assessment of discriminant validity was to evaluate the cross-loading values of reflective constructs' items (Table 10). The constructs of the reflective measurement model were tested for reliability and validity in this study. For formative measurement models, convergent validity, collinearity, and outer weights were assessed. Table 6 depicts the important value criteria for the acceptability of reflective and formative measurement approaches.

**Table 8.** Discriminant Validity (Fornell–Larcker Criterion).

| Constructs | ATT | INT | RA |
|---|---|---|---|
| ATT | 0.740 | | |
| INT | 0.699 | 0.760 | |
| RA | 0.633 | 0.666 | 0.723 |

**Table 9.** Heterotrait–Monotrait Ratio (HTMT).

| Constructs | ATT | INT |
|---|---|---|
| INT | 0.698 | |
| RA | 0.729 | 0.814 |

**Table 10.** Cross loadings among reflective scale measurements.

| | ATT | INT | RA |
|---|---|---|---|
| Att 1 | 0.690 | 0.476 | 0.473 |
| Att 2 | 0.779 | 0.573 | 0.571 |
| Att 3 | 0.748 | 0.500 | 0.579 |
| Int 1 | 0.501 | 0.716 | 0.605 |
| Int 2 | 0.528 | 0.755 | 0.607 |
| Int 3 | 0.528 | 0.755 | 0.714 |
| Int 4 | 0.536 | 0.766 | 0.581 |
| Int 5 | 0.562 | 0.804 | 0.663 |
| RA 1 | 0.591 | 0.700 | 0.806 |
| RA 2 | 0.481 | 0.532 | 0.656 |
| RA 3 | 0.451 | 0.600 | 0.615 |
| RA 4 | 0.583 | 0.666 | 0.796 |

*4.3. Assessment of Formative Measurement Models*

The evaluation process of a formative measurement model is distinct from a reflective measurement model [116,117,120]. In the formative measurement model, every item of a latent construct represents an independent cause. These items were not highly correlated. In the case of assessing the convergent validity of a formative construct, redundancy analysis was carried out for each latent variable separately. Existing latent variables (formative) are considered exogenous variables to predict the endogenous variable, which is calculated by one or more reflective measurement items [121]. Figure 2 shows the global item (Global_LC level), which stands for the overall essence of all indicators of Low Complexity level (LC 1, LC 2, and LC 3). Figure 3 depicts the global COMP indicator, which is a global item that encapsulates the notion of formative compatibility indicators (COMP 1, COMP 2, and COMP 3). In order to calculate this, two new models were developed in SmartPLS, as shown in Figures 2 and 3, as we had two formative measurement models. If the correlation coefficient between latent variables is greater than or equal to 0.80, convergent validity is

established [116,117,121]. Figures 2 and 3 show that the correlation coefficients between latent variables for the two formative constructs (Lower Complexity level = 0.80 and Compatibility = 0.90) reached the threshold value of 0.80. Hence, formative measurement models had convergent validity.

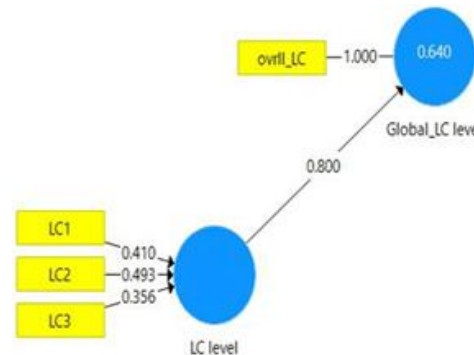

**Figure 2.** Redundancy analysis (LC level).

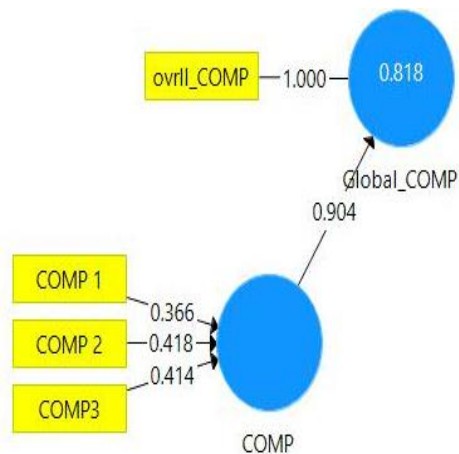

**Figure 3.** Redundancy analysis (Compatibility).

A problem of collinearity may occur if the items or indicators of a formative measurement model are highly correlated with each other. To assess the collinearity status, the variance inflation factor (VIF) value was estimated and is presented in Table 11. This showed that the VIF values were lower than the threshold value of 3.3 [118], suggesting that all items of their underlying construct were distinct and measured different aspects of that construct. The outer weights of formative construct items were calculated to determine the relative importance of those items in terms of their underlying latent construct. In defining the construct, the significance of each item of the formative construct's outer weight was assessed. Table 12 shows the bootstrapping results of 5000 sub-samples and reveals the outer weights and path coefficients of items in the construct. The results showed that all items in the formative construct had weights that were both positive and significant. This means that all items in the formative construct met the criteria of relevance and importance.

**Table 11.** Collinearity assessment.

| Item | VIF |
|---|---|
| LC1 | 1.336 |
| LC2 | 1.372 |
| LC3 | 1.554 |
| COMP1 | 1.712 |
| COMP2 | 1.723 |
| COMP3 | 1.507 |

**Table 12.** Assessment of formative model.

| Items | Outer Weight | Std. Error | T-Value | *p*-Value |
|---|---|---|---|---|
| LC1 -> LC level | 0.502 ** | 0.112 | 4.495 | 0.000 |
| LC2 -> LC level | 0.275 * | 0.120 | 2.283 | 0.022 |
| LC3 -> LC level | 0.466 ** | 0.110 | 4.245 | 0.000 |
| COMP 1 -> COMP | 0.337 * | 0.118 | 2.862 | 0.004 |
| COMP 2 -> COMP | 0.359 * | 0.113 | 3.192 | 0.001 |
| COMP 3 -> COMP | 0.500 ** | 0.113 | 4.439 | 0.000 |

Note: * and ** indicates the significance level at 5% and 1%, respectively.

### 4.4. Assessment of the Structural Model

Before evaluating the structural model, the multicollinearity in the inner model was checked (Table 13). The results showed that there was no multicollinearity problem in the model with VIF values less than 3.3.

**Table 13.** Collinearity assessment (inner VIF).

| Construct | ATT | INT |
|---|---|---|
| ATT | | 1.000 |
| LC | 2.531 | |
| RA | 3.218 | |
| COMP | 3.011 | |

The assessment of the structural model was performed to examine the relationship between the latent variables using four criteria: coefficient of determination ($R^2$), effect size ($f^2$), predictive relevance ($Q^2$) and path coefficients (Table 12). Hypothesis testing was performed once the latent variables were tested and confirmed regarding reliability and validity. Then, a bootstrapping procedure with 5000 sub-samples was performed to evaluate the significance of the path coefficients. The guidelines used to evaluate the structural model are summarized in Table 14.

**Table 14.** Assessment of the structural model.

| Criterion | Guideline |
|---|---|
| Coefficient of determination ($R^2$) | 0.25—Weak<br>0.50—Moderate<br>0.75—Substantial |
| Path Coefficient | between −1 and +1 |
| Effect Size ($f^2$) | 0.02—Small effect<br>0.15—Medium effect<br>0.35—Large effect |
| Predictive relevance ($Q^2$) | Above zero |

Source: [86,122].

The coefficient of determination ($R^2$) explained the predictive accuracy of the model, which assessed the overall quality of the PLS model. The value indicates how much

of the variance in an endogenous variable can be explained by an exogenous (latent construct) variable. Table 15 shows that the value of $R^2$ was 0.489 for intention and 0.592 for Attitude. These values indicate that the proposed models had moderate explanatory power. Hair et al. [116] indicated that a model that only accesses $R^2$ values is not trustworthy.

**Table 15.** Summary of Effect Size ($f^2$), Predictive Relevance ($Q^2$), and Coefficient of Determination ($R^2$).

| Constructs | ATT ($f^2$) | INT ($f^2$) | Effect Size | Predictive Relevance ($Q^2$) | R Square ($R^2$) |
|---|---|---|---|---|---|
| INT | | | | 0.433 | 0.489 |
| ATT | | 0.956 | large | 0.420 | 0.592 |
| LC | 0.049 | | small | | |
| RA | 0.163 | | medium | | |
| COMP | 0.079 | | small | | |

Therefore, to evaluate the predictive relevance of the structural model, Stone [123] introduced $Q^2$. Latent exogenous constructs in the structural model have predictive relevance if the value of $Q^2$ is greater than zero [116,120]. $Q^2$ values of 0.43 and 0.420 were both higher than zero, which means that endogenous constructs had enough predictive relevance.

The effect size ($f^2$) identifies how much an exogenous variable contributes to an endogenous variable value of $R^2$ [116]. The results revealed that Attitude was the best predictor, as it had the largest effect on the continuance intention accumulated by Relative Advantage ($f^2 = 0.163$), Compatibility ($f^2 = 0.079$), and Low Complexity levels ($f^2 = 0.049$).

The results of the path coefficient using the bootstrapping procedure with sub-samples of 5000 cases for the hypothesized relationships are presented in Table 16. The results showed that RA ($\beta = 0.337$; $t = 2.325$; $p = 0.020$) had the highest significant positive impact on Attitude, which supports $H_1$. Equivalently, the relationship between the LC level and Attitude ($\beta = 0.225$; $t = 2.256$; $p = 0.024$) was positive and significant, supporting the second hypothesis ($H_2$). $H_3$ was also supported because Compatibility had a positive and significant influence on Attitude ($\beta = 0.273$; $t = 2.085$; $p = 0.037$). Therefore, all relationships were significant and supported the hypotheses presented in Table 16. The positive and significant relationship between Attitude and Intention ($\beta = 0.699$; $t = 9.878$; $p = 0.000$) validated $H_4$.

**Table 16.** Path coefficient and hypothesis testing.

| Relationship | Std. $\beta$ | Std. Error | $t$-Value | $p$-Value | Decision |
|---|---|---|---|---|---|
| ATT -> INT | 0.699 ** | 0.071 | 9.878 | 0.000 | Supported |
| LLC -> ATT | 0.225 * | 0.100 | 2.256 | 0.024 | Supported |
| RA -> ATT | 0.337 * | 0.145 | 2.325 | 0.020 | Supported |
| COMP -> ATT | 0.273 * | 0.131 | 2.085 | 0.037 | Supported |

Note: * and ** denote a 5% and 1% significance level, respectively.

The estimated original TAM structural model was statistically fit and well adjusted to the data (Figure 4). The coefficient of determination ($R^2$) in the original TAM was 0.545 for Attitude and 0.458 for behavioral intention. The integration of TAM and DOI theory is presented in Figure 5. In this integrated model, the values of the coefficients of determination for both exogenous and endogenous constructs increased (from 0.545 to 0.592 for attitude and from 0.458 to 0.489 for intention). Of the variance in attitude, 59.2% was explained by the included constructs, while 48.9% of the variance in the behavioral intention toward CA was explained by the constructs considered in the model.

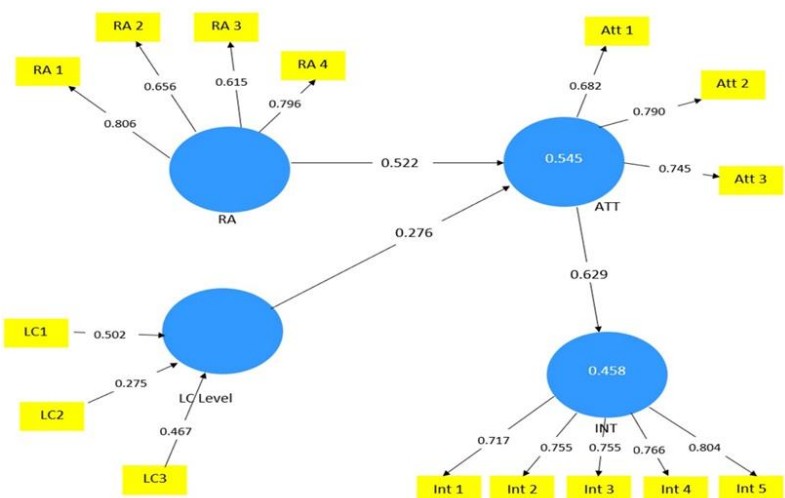

**Figure 4.** TAM framework considered in the study.

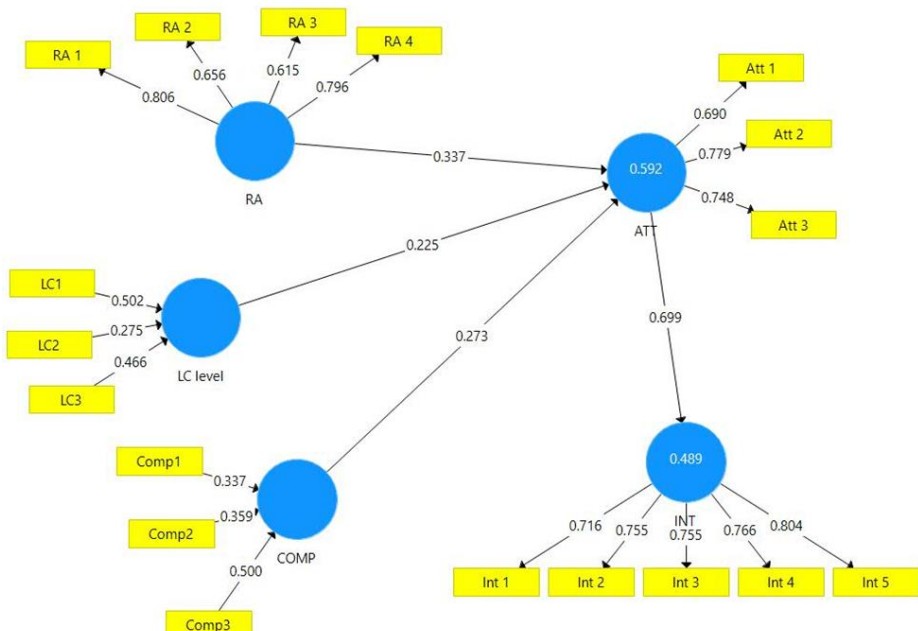

**Figure 5.** Integrated framework of TAM and DOI theory.

## 5. Conclusions and Discussion

### 5.1. Conclusions

CA has immense potential for increasing agricultural productivity, ensuring food security, and achieving self-sufficiency in food production. In addition, it is a way forward to generate socio-economic well-being and, aside from that, maintain a sustainable environment. What is more important is that CA practice can be a fortified tool against the threats of global climate change [124]. As a significant portion of the total population, developing countries such as Bangladesh are at risk of losing their means as a result of a climatic catastrophe combined with concurrent environmental deterioration resulting from rapid urbanization and unplanned development programs. Hence, it is vital for this country to understand the invaluable emergence of CA practices to tackle further climatic and food insecurity threats. Keeping all these facts in mind, the study devoted itself to investigating factors affecting farmers' attitudes toward the intention of CA practice using the PLS-SEM approach. The literature indicates that DOI and TAM are effective socio-psychological frameworks that can help researchers explain factors that affect farmers' intentions toward

their behavior [38,49,51]. However, it is notable that recent conservation agriculture studies [14,18,20,22,30,125–127] did not integrate these two theories in understanding farmers' attitudes directed toward the intention of CA. Therefore, the findings discussed below make a novel contribution to the existing body of literature.

Relative Advantage (RA) is usually the strongest predictor of the adoption of any new technology [60]. This study's findings also show that RA ($\beta = 0.337$ and significant at $p < 0.05$) had the least direct significant positive impact on Attitude toward CA, supporting the first hypothesis ($H_1$). This is consistent with previous findings that RA (Perceived usefulness) has a significant positive impact on Attitude and technology adoption [20,32,81,88,109]. The results also reveal that CA has several advantages compared to existing CF practices. In addition, it indicates that the adoption of CA is likely to expand more rapidly in the following years if the adopters' feel that the relative advantage of CA is higher than the existing practices. However, it should be emphasized that adopters (such as innovators, early adopters, and middle/late adopters) were not further classified in this study.

Low Complexity level ($\beta = 0.225$ and significant at $p < 0.05$) had a significant direct positive impact on Attitude and an indirect positive impact on continuance intention toward CA, supporting the second hypothesis ($H_2$). Studies [44,75,128–130] have shown that complexity (perceived ease of usefulness) has a positive and significant relationship with Attitude. However, Adnan et al. [131] conducted a similar study among paddy farmers and found this relationship to be positive but not significant. When an individual finds a technology easier to operate, there will be a positive perception of its adoption and usage in the future [132]. This result does not fully agree with Corrigan [133] and Jamshidi and Hussin [85], who found a positive but non-significant relationship between complexity and intention.

Compatibility ($\beta = 0.273$ and significant at $p < 0.05$) has a significant direct positive impact on attitude and an indirect impact on continuance intention toward CA, supporting the third hypothesis ($H_3$), which is consistent with Jamshidi and Hussin [85]. Wu et al. [134] and Sharifzadeh et al. [135] argued that when an individual finds a technology consistent with his or her prior experience, values, and work conditions, he or she will feel more confident in adopting it and have a higher perception of this technology advantage. However, a few studies [136,137] found that the attributes of innovation (i.e., Relative Advantage, Complexity, and Compatibility) and Attitude are incompatible. In contrast, Ayodele et al. [138] illustrated a positive and significant relationship between compatibility and behavioral intention. According to Ducey and Coovert [129], a technology that assists an individual with his or her work would give him or her belief in the usefulness of the technology and help improve his or her performance.

According to the findings, the construct attitude ($\beta = 0.699$ and significant at $p < 0.001$) has the greatest significantly positive impact on CA continuance intention, supporting the fourth hypothesis ($H_4$). This result is supported by studies conducted in various disciplines on predicting behavioral intention [58,131,139–141]. However, several previous studies have shown that Attitude has a significantly positive but not the highest impact on behavior [44,75,128,142,143]. When people have a favorable attitude, they are mentally ready to make more accurate decisions [144].

### 5.2. Discussion

The results have several practical implications for researchers, practitioners, and policymakers. The findings show that there is a significant and positive relationship between RA and Attitude toward CA farming in Bangladesh. Moreover, RA had the highest impact on Attitude compared to other constructs, such as Compatibility and LC level. Extension agents, the government, and non-government organizations can train farmers on the benefits of CA over existing conventional farming practices to speed up adoption. A growing number of programs, campaigns, and policies must aim to improve farmers' knowledge and Attitude. Local government organizations can provide incentives not only to farmer organizations but also to private entities that can organize events to

display and inform the public about the relative advantage and compatibility of CA. These programs can better inform farmers about the low complexity level and change their perceptions. Linking various government and non-government sectors has worked to promote CA adoption in some other contexts, including Bangladesh [23,145,146]. These events can raise farmers' awareness and positively transform their perceptions of CA. Eventually, these will enhance the attitudes of the farmers toward the continuation of CA farming, which will translate into their actual behavior. Second, a farmer's intention is largely dependent on several other factors. In this model, Attitude and farmers' intentions are strongly positively related. Public institutions must take the lead in assisting CA farmers by developing and implementing appropriate incentive and subsidy programs that take into account the constructs identified in this study as relevant.

This research work integrated the Diffusion of Innovation (DOI) theory and the Technology Acceptance Model (TAM) validated in the context of CA, providing more insights into the farmers' potential perceptions about their continuance intention toward CA farming practices. The findings highlight the key propositions of the proposed model (integrating TAM and DOI) and the applicability of this model to more effectively predict intentions through attitude. Relative Advantage (RA), Compatibility, and Low Complexity (LC) levels had a positive and statistically significant effect on the attitudes of farmers toward CA farming. However, RA had a greater influence on Attitude than the other construct. As a result, policy emphasis should be placed on the perceptions of the (relative) advantages of CA to accelerate a positive attitude and thus influence farmers' behavioral intentions to continue CA farming practices. To speed up the sustainable adoption of CA practices in Bangladesh, improved perceptions of RA and a positive attitude must be enhanced. Government institutions, as well as private institutions and NGOs, should arrange training programs regularly in every upazila so that farmers can easily go and receive training on CA. Field Day is an important extension tool through which many farmers can gain updated knowledge and information. The use of ICT-based platforms, applications, and software can be promoted, especially since Bangladesh has significantly developed its ICT-related infrastructure at the union level in recent years. Every union has a Union Digital Center (UDC), which can be utilized to enhance access to information and services related to CA in the respective localities.

*5.3. Limitation*

Despite being a novel effort regarding CA practice in Bangladesh, which accommodates factors affecting farmers' attitudes and behavioral intentions toward CA, the study's scope is nevertheless constrained by specific limitations. First, the sample size in future studies can be larger, which will help to better understand the wider picture of CA practice in Bangladesh. Second, this study did not include farmers who had never adopted CA or those who had left after adoption. Further studies could be directed toward scaling the impact of CA practices on the food security status of small-scale farmers and those who were not included in this research. In addition, studies that attempt to highlight the impact of CA on the livelihood and income status of farmers are highly anticipated. Lastly, prospective studies can adopt more advanced statistical tools and machine learning approaches in their analysis of CA practices for dynamic visualization.

**Author Contributions:** Conceptualization, R.A.Z.T., M.M.H. and Y.L.; methodology, R.A.Z.T., M.M.H., M.Y. and M.J.A.; software, R.A.Z.T., M.M.H. and Y.L.; validation, M.Y. and M.J.A.; formal analysis, R.A.Z.T., M.M.H. and M.Y.; investigation, R.A.Z.T. and M.M.H.; resources, Y.L. and M.Y.; data curation, R.A.Z.T. and Y.L.; writing the original draft preparation, R.A.Z.T., M.M.H., M.J.A., M.Y. and Y.L.; writing, review, and editing, R.A.Z.T., M.M.H., M.Y. and Y.L.; visualization, R.A.Z.T., M.Y. and M.M.H.; supervision, Y.L. and M.Y.; project administration, R.A.Z.T., M.Y. and Y.L.; funding acquisition, M.Y. and Y.L. All authors have read and agreed to the published version of the manuscript.

**Funding:** This research was supported by the "Talent Project of North China University of Technology" (Program No. 20210115).

**Institutional Review Board Statement:** Not applicable.

**Data Availability Statement:** The datasets used and analyzed during the current study are available from the corresponding author on reasonable request.

**Acknowledgments:** We are thankful to the Regional Agricultural Research Station (RARS) and the Regional Wheat Research Center (RWRC) of Bangladesh for their assistance with data collection.

**Conflicts of Interest:** The authors declare no conflict of interest.

## Appendix A

**Table A1.** Result of Endogeneity Test using the Gaussian Copula Approach.

| Test | Construct | Coefficient | *p* Value |
|---|---|---|---|
| Gaussian copula of model 1 | RA | 0.332 | 0.000 |
| | LC | 0.215 | 0.000 |
| | COMP | 0.271 | 0.030 |
| | ATT | 0.674 | 0.000 |
| | RA [c] | 0.037 | 0.712 |
| Gaussian copula of model 2 | RA | 0.331 | 0.000 |
| | LC | 0.239 | 0.000 |
| | COMP | 0.269 | 0.031 |
| | ATT | 0.683 | 0.000 |
| | LC [c] | 0.027 | 0.598 |
| Gaussian copula of model 3 | RA | 0.328 | 0.000 |
| | LC | 0.233 | 0.000 |
| | COMP | 0.266 | 0.037 |
| | ATT | 0.691 | 0.000 |
| | COMP [c] | 0.052 | 0.413 |
| Gaussian copula of model 4 | RA | 0.319 | 0.000 |
| | LC | 0.217 | 0.000 |
| | COMP | 0.279 | 0.029 |
| | ATT | 0.659 | 0.000 |
| | ATT [c] | 0.016 | 0.822 |
| Gaussian copula of model 5 | RA | 0.330 | 0.000 |
| | LC | 0.219 | 0.000 |
| | COMP | 0.274 | 0.029 |
| | ATT | 0.686 | 0.000 |
| | RA [c] | 0.039 | 0.813 |
| | LC [c] | 0.026 | 0.579 |
| Gaussian copula of model 6 | RA | 0.329 | 0.000 |
| | LC | 0.207 | 0.000 |
| | COMP | 0.263 | 0.033 |
| | ATT | 0.690 | 0.000 |
| | RA [c] | 0.031 | 0.789 |
| | COMP [c] | 0.051 | 0.411 |
| Gaussian copula of model 7 | RA | 0.318 | 0.000 |
| | LC | 0.221 | 0.000 |
| | COMP | 0.231 | 0.031 |
| | ATT | 0.663 | 0.000 |

Note: [c] indicates the copula term in the model.

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
