# Peer review of "An Application of Partial Least Squares Structural Equation Modeling (PLS-SEM) to Examining Farmers’ Behavioral Attitude and Intention towards Conservation Agriculture in Bangladesh"

_agriculture, doi:10.3390/agriculture13020503_

Round 1
Reviewer 1 Report
1. The starting point for this manuscript is the benefits of conservation agriculture (CA). For example, on lines 52-54, the authors state, "In the epoch of modernization and technological advancement, when climate change has become a major global development concern, conservation agriculture (CA) can play a pivotal role in ensuring food supply while preserving the earth." However, CA as defined by the authors (minimum or no till, soil cover, diversification) does not always increase crop yields or net farm income. Two recent reviews of the literature on reduced tillage make this point (Cameron M. Pittelkow, "When does no-till yield more? A global meta-analysis," Field Crops Research, Volume 183, November 2015, 156-168; Macson O. Ogieriakhi and Richard T. Woodward, "Understanding why farmers adopt soil conservation tillage: A systematic review," Soil Security, Volume 9, December 2022, 100077).
The bottom line is that non-adopters of CA in Bangladesh may have sound agronomic and economic reasons for their decision not to adopt.
2. Based on information cited by the authors and another recent study that the authors do not cite (Shaheen Akter et al., "Adoption of conservation agriculture-based tillage practices in the rice-maize systems in Bangladesh," World Development Perspectives, Volume 21, March 2021, 100297), it appears that adoption of CA in Bangladesh is very low. For example, Akter et al. find an overall adoption rate of 6.6% in the rice-maize systems they studied.
In the literature on technology adoption and diffusion, that would put the farmers in Bangladesh practicing CA, including the farmers in the authors’ sample, in the categories of "innovators" and "early adopters". A recurring theme in the technology adoption literature is that innovators and early adopters have different personal characteristics from middle/late adopters such as novelty-seeking behavior, attitudes toward risk, and use of external sources of information. Innovators and early adopters also tend to be younger and more educated than the middle/late adopters.
This raises a concern about external validity. The results that the authors find might generalize to other innovators and early adopters, but probably not to the middle or late adopters. For example, on lines 527-529, the authors state, "the adoption of CA is likely to expand more rapidly in the following years if the adopters’ feel that the relative advantage of CA is higher than the existing practices." I think the scope of that statement is limited to the early adopters who have not yet adopted.
3. The authors frame their research questions in terms of adoption and continuance of CA (lines 79-80). My comment #1 focuses on adoption, here my comment is about continuance. The authors ask (lines 81-82), "Why are CA farmers not keen to continue CA farming practices?" This would suggest that either (1) there is churn in the population of farmers using CA, with some starting each year and some quitting, or (2) farmers who initially adopted CA are gradually quitting and the adoption rate is on a downward trend. However, the manuscript does not present any statistics on churn or trends over time in adoption. Some statistics are necessary to convince the reader that continuance is an issue worthy of study.
4. In discussing their manuscript’s contribution to the literature, the authors state on lines 120-122, "Second, the study has shown whether, in a developing country like Bangladesh, the TAM theory and the DOI theory for any new technological adoption hold true or not." Why is this a contribution to the literature? Did a prior study find that the theories do not work well in developing countries? Why should a country’s status as developing or developed matter as far as these theories are concerned?
5. In section 5.2 (lines 568-573), the authors state, "Local government organizations can provide incentives not only to farmer organizations but also to private entities that can organize events to display and inform the public about the relative advantage and compatibility of CA. These programs can better inform farmers about the low complexity level and change their perception. Eventually, these will enhance the attitude of the farmers towards the continuation of CA farming, which will translate into their actual behavior." Are there studies that have found that this approach actually works? For this approach to work, farmers have to (1) attend these events, (2) believe what they are being told, and (3) be open to change in their perceptions.
Reviewer 2 Report
This paper applied the Technology Acceptance Model (TAM) and Diffusion of Innovation (DOI) theories to examine the socio-psychological determinants of Bangladeshi farmers’ behavior regarding the adoption of CA. Based on the data collected from 201 CA farmers, the author used a variance-based structural equation modeling (PLS-SEM) approach to test the model. The results show that the model does have better explanatory power.
Introduction: The introduction is well written. I have no points to add.
Conceptual Framework is well written.
Material and Methods are also well written.
Additional Questions:
1. It is suggested that the author optimizes the table format in this paper.
2. In the results section, the author needs to be careful to explain the economic implications.
3. Figure 2 is a little fuzzy
Conceptual framework: The author should be explicit about his opinion. Why should the author explain attitude from three dimensions?
Reviewer 3 Report
In this paper, the factors influencing the behavioral attitude and intention of farmers toward conservative agriculture were examined using partial least squares structural equation modeling (PLS-SEM). With the use of statistical package software such as Smart-PLS and R, it is easy for anyone to perform computations using PLS-SEM. For this reason, many papers using PLS-SEM have been published in social science journals, especially in marketing research. When the sample size is small, as in this study, it seems appropriate to use PLS-SEM instead of general covariance-based SEM.
1) Only farmers currently practicing conservation agriculture were included in the study. Excluding farmers who had stopped practicing conservation agriculture from the survey would have biased the estimation results of the analysis. A more in-depth discussion could have been developed by asking farmers who had stopped conservation agriculture why they had stopped.
2) The paper points out that the relative advantage (RA) of conservation agriculture has the greatest influence on farmers' attitudes towards continuing conservation agriculture. However, it did not analyze the factors and farmer characteristics that influence farmers' perceptions of relative advantage. Although I understand that it is necessary to check whether the estimation model meets many validity criteria, the more important point is to clarify which farmers are more likely to continue conservation agriculture and which farmers are not. The point that the relative advantage (RA) of conservation agriculture has the greatest influence on farmers' attitudes towards continuing conservation agriculture is a given and does not sound new to me.
3) Although what the authors are trying to explain is clear, some paragraphs are so long that they are somewhat difficult to read.
Round 2
Reviewer 1 Report
The authors' responses and revisions to their manuscript address my concerns.
Author Response
We cannot find anything that needs to be revised. All of the changes have been accepted by the reviewer.
Reviewer 3 Report
All potential enhancements and corrections have been done based on the feedback. Consequently, I deem it suitable for publication in its current form.
Author Response
We cannot find anything that needs to be revised. The reviewer has accepted all of the changes.
Reviewer 3 appears to have mistakenly chosen "not signed the review report."